# A Survey of Wild Indigenous *Cryptostylis ovata* Orchid Populations in Western Australia Reveals Spillover of Exotic Viruses

**DOI:** 10.3390/v17010108

**Published:** 2025-01-14

**Authors:** Stephen Wylie, Hua Li, Shu Hui Koh

**Affiliations:** 1Food Futures Institute, Murdoch University, 90 South Street, Perth 6150, Australia; perthmuzi@yahoo.com; 2School of Medical, Molecular and Forensic Sciences, College of Environmental and Life Sciences, Murdoch University, 90 South Street, Perth 6150, Australia; s.koh@murdoch.edu.au

**Keywords:** virus emergence, wild plant virology, virus transmission, potyvirus

## Abstract

*Cryptostylis ovata* is a terrestrial orchid endemic to southwestern Australia. The virus status of *C. ovata* has not been studied. Eighty-three *C. ovata* samples from 16 populations were collected, and sequencing was used to identify RNA viruses from them. In one population, all tested plants were co-infected with isolates of the exotic-to-Australia viruses Ornithogalum mosaic virus (OrMV) and bean yellow mosaic virus (BYMV). In another population, one plant was infected with BYMV. No viruses were detected in the remaining populations. The OrMV isolate shared 98–99% nucleotide identity with isolates identified from wild indigenous *Lachenalia* (Iridaceae) plants in South Africa. This suggests that the source of OrMV in *C. ovata* may be one or more bulbous iridaceous flowering plants of southern African origin that were introduced to Western Australia as ornamentals and that have since become invasive weeds. One BYMV isolate from *C. ovata* also exhibited 99% nucleotide identity with strains isolated from the exotic leguminous crop *Lupinus angustifolius* in Western Australia, suggesting possible spillover to indigenous species from this source. This study with *C. ovata* highlights the probable role of invasive weeds and exotic crops as sources of exotic virus spillovers to indigenous plants.

## 1. Introduction

Australia is a global centre of orchid biodiversity. Of the 29,524 orchid taxa accepted internationally [1], approximately 1700 are indigenous to Australia [2]. Wild orchids face numerous threats, primarily habitat loss due to human activity, including climate change. Specific threats include plant removal for collections, the decline of symbiotic pollinator species and mycorrhizal fungi, destruction by animals, competition from invasive weeds, and damage caused by pests and diseases [3,4,5].

The genus *Cryptostylis* comprises 23 species of orchids with a predominantly Southern Hemisphere distribution. The greatest number of species occur in the Philippines, Indonesia, and Papua New Guinea region (World Checklist of Vascular Plants, Royal Botanic Gardens, Kew, 2024). The genetic analysis of *Cryptostylis* species from Australia and Asia supports an Australian origin for the genus, followed by a single dispersal event to Asia and subsequent speciation [6]. Of the five *Cryptostylis* species indigenous to Australasia, four are found in eastern Australia and New Zealand, while one, *Cryptostylis ovata*, is confined to southern Western Australia [7]. Notably, *C. ovata* is the only evergreen orchid among the approximately 400 terrestrial orchid taxa indigenous to Western Australia; all others are deciduous, surviving much of the year as underground bulbs or tubers.

Viruses infecting members of Orchidaceae in Australia, as with the flora generally, are poorly studied. While many virus species identified from Australian orchids appear to have evolved on the continent, only *Divavirus* and *Platypuvirus* represent genera that may be endemic to Australia. Most other apparently indigenous viruses identified from Australian orchids belong to internationally distributed genera, such as *Potyvirus*. Examples include *Diuris virus Y* [8] (*Potyvirus*), *Donkey orchid virus A* [9] (*Potyvirus*), *Caladenia virus A* [9] (*Poacevirus*), and *Pterostylis blotch virus* [10] (*Orthotospovirus*).

In addition to indigenous viruses, several Australian orchids are infected with exotic viruses. These viruses, which originate outside Australia, were likely introduced over the past two centuries following British colonisation through plants imported for agriculture, horticulture, and landscaping, and from weeds, with subsequent spillover to wild plants, including native orchids. For example, isolates of bean yellow mosaic virus (BYMV, *Potyvirus*) and Ornithogalum mosaic virus (OrMV, *Potyvirus*) were identified in potted *Diuris magnifica* orchids in Perth, W.A., and in wild *D. corymbosa* orchids in remnant bushland near Brookton, Western Australia [11]. In eastern Australia, BYMV has been detected in *Pterostylis curta* and *Diuris* species orchids, while OrMV (referred to as *Pterostylis virus Y* but later confirmed to be OrMV) was identified from species of *Pterostylis*, *Chiloglottis*, *Diuris*, *Eriochilus*, and *Corybas* orchids [8].

To date, no *Cryptostylis* species has been investigated for the presence of viruses. In this study, we tested leaf samples from wild populations of *C. ovata* for RNA viruses using a high-throughput sequencing approach. This method is advantageous over other virus detection techniques such as PCR- and antibody-based approaches as it does not require prior knowledge of viruses present.

## 2. Materials and Methods

Leaf samples were collected from 83 individual *C. ovata* plants from 16 populations in southern Western Australia (Table 1). Due to the lateral spread of *C. ovata* plants through rhizomes, it was challenging to delineate individual plants. To address this, leaves spaced at least 1 m apart were considered to belong to individual plants. Sampling sites included a variety of habitats, ranging from species-depleted small remnant bushlands on roadsides, large indigenous woodlands of diverse mixed species, to the species-sparse monocultures of exotic *Pinus radiata* plantations. The aim was to collect samples from 10 individuals per population; however, in populations with fewer than 10 plants, one sample was collected from every available plant. Samples were taken regardless of visible symptoms of virus infection, such as chlorosis on young leaves, mosaic patterns, necrosis, or stunting.

Total RNA was extracted from 100 mg of leaf tissue using the RNeasy Plant Mini Kit (Qiagen, Hilden, Germany) after grinding the samples in liquid nitrogen. To prepare for cDNA library construction, ribosomal RNA was depleted using the Ribo-Zero Plant kit, and libraries were generated with the TruSeq Stranded Total RNA Plant Library preparation kit (Illumina, San Diego, CA, USA). Paired-end high-throughput sequencing (HTS) was performed on an Illumina NovaSeq 6000 S4 platform with 150 cycles.

Post-sequencing, TruSeq primer-adaptors were removed, and quality trimming was performed using default parameters in CLC Genomics Workbench (Qiagen). Reads shorter than 100 nucleotides were discarded. De novo assembly was conducted in both CLC Genomics Workbench and Geneious Prime (Biomatters, New York, NY, USA). Contigs longer than 100 nucleotides were analysed using Blastn. Results were screened for matches with viruses and viroids. Contigs matching these sequences and those with no matches (referred to as orphans) were subjected to further analysis.

Orphan contigs were translated in six reading frames (three forward and three reverse). Contigs without open reading frames (ORFs) of at least 100 amino acid residues were discarded. For those with ORFs, nucleotide and amino acid sequences were analysed in Blastn or Blastp, respectively, for similarities to known sequences.

Based on HTS results, five sets of species-specific primers were synthesised for each identified virus (Appendix A). Primers were designed in Primer 3, each with a melting temperature of 60 °C. Amplification cycle conditions were as follows: melting at 95 °C 10 s, annealing at 60 °C 10 s, and elongation at 70 °C for 20 s for 25 cycles.

All 83 RNA samples were then tested using these primers via RT-PCR. Amplified bands were sequenced using the Sanger method. Primer sequences were trimmed from the resulting sequences, which were then aligned against HTS-derived sequences and publicly available sequences from GenBank using Blastn. Gaps between amplicons were filled by combining appropriate primers and performing additional Sanger sequencing. This process enabled the identification of virus-derived sequences and facilitated the assembly of complete or partial genome sequences for each virus present in infected plants.

Phylogenetic analysis was performed on genome sequences of viruses after alignment in ClustalW. A maximum liklihood tree using the Tamura–Nei substitution model was constructed within Mega11 and the bootstrap method with 1000 replications was used as the test of phylogeny.

## 3. Results

### 3.1. Plant Samples

In each population except the Bowelling-Duranillin Road population, the plants appeared to be healthy, lacking symptoms typical of virus infection. In the Bowelling-Duranillin Road population, all eight plants displayed mosaic patterns and chlorotic streaking symptoms reminiscent of virus infection (Figure 1).

### 3.2. Viruses

Following RNA sequencing, viruses were identified from two *C. ovata* populations. The eight samples tested from the Bowelling-Duranillin Road population were doubly infected with bean yellow mosaic virus (BYMV) and Ornithogalum mosaic virus (OrMV). One of the six plants from the Devlin Rd population was infected with an isolate of BYMV. No viruses were identified from the other *C. ovata* plants tested.

### 3.3. Sequence Analysis

The alignment of the eight BYMV sequences from the Bowelling-Duranillin Road population revealed that they were identical. This isolate was named BYMV-BDW. One of the samples collected from the Devlin Rd population was infected with a distinct isolate of BYMV (Figure 2).

The eight OrMV sequences from the Bowelling-Duranillin Road population were identical, and this isolate was named OrMV-BDW (Figure 3).

The complete genome sequence of BYMV-BDW (Genbank accession PQ213087) was 9481 nt, encoding 3056 aa, while isolate BYMV-Devlin (PQ285386) was incomplete, at 8387 nt at the 5′ end, encoding 2749 aa and lacking the 5′ untranslated region and the terminal region of the P1 gene. Isolates BYMV-BDW and BYMV-Devlin were not identical, sharing 91% nt and 96% aa identities over the common regions. The phylogenetic analysis of the nucleotide sequences of these two isolates with other BYMV genome sequences available from GenBank revealed that BYMV-Devlin shared 99% identity with four other BYMV isolates from Western Australia, two from wild symptomatic *Diuris* orchids and two from *Lupinis angustifolius* (narrow-leafed lupin) plants collected from crops. In contrast, BYMV-BDW was not closely aligned with the other BYMV isolates from Australia, instead sharing up to 92% identity with BYMV isolates from India, Taiwan, Japan, and the USA, collected from diverse hosts (Figure 2).

The complete genome sequence of isolate OrMV-BDW was 9045 nt (PQ213088). OrMV-BDW shared greatest identities (98–99%) with OrMV isolates identified from indigenous *Diuris* orchids from Western Australia, and also with an isolate from a wild *Lachenalia* sp. plant collected in South Africa, where it is an indigenous species (Figure 3).

## 4. Discussion

This paper presents the first report of viruses infecting plants of the indigenous Australian orchid *Cryptostylis ovata.* Of the sixteen *C. ovata* populations tested, only two populations had plants infected with one or two viruses. The two viruses, BYMV and OrMV, have origins outside Australia. No indigenous viruses were identified.

Interestingly, the eight plants collected from the Bowelling-Duranillin Road population were all infected with identical isolates of BYMV and OrMV. Thus, it seems likely that this ‘population’ comprises a single large plant that has spread several metres in diameter and in which the virus has moved systemically.

BYMV has a broad international distribution and host range, infecting both monocotyledonous and dicotyledonous plants in all temperate cropping regions. It poses a serious threat to leguminous crops, such as narrow-leafed lupins and clover/medic pastures in Western Australia [12,13] and elsewhere, as well as to floriculture crops such as *Gladiolus* sp. in several countries [14,15]. BYMV has been reported from a range of wild and cultivated orchids, including *Vanilla* and other orchids in French Polynesia [16], *Masdevallia* orchids from the USA, and *Calanthe* orchids from Japan [17]. Like other potyviruses, BYMV is transmitted horizontally by aphids, mechanically by physical contact and via pollen, and vertically through seeds in several plant species [18,19], although seed transmission has not been recorded in orchids.

No aphids are reported infesting the leaves of *C. ovata*, and, indeed, these authors have not observed this in decades of observation. However, the authors have often observed aphids feeding on *C. ovata* flowers, suggesting this may be the means through which aphids transmit viruses. Notably, *C. ovata* produces flowers in mid-summer, after some major BYMV sources, such as narrow-leafed lupin crops, have already been harvested in Western Australia.

BYMV is a genetically diverse virus, with several groupings proposed based on nucleotide sequence phylogeny and host preferences [20,21,22]. BYMV-Devlin shares the greatest genetic identity with isolate BYMV-SW3.2 already described from Western Australia in wild orchids (JX156423) (Figure 2), suggesting this strain is widespread among different hosts in the region; these two host orchids were located approximately 140 km apart. In the *Diuris corymbosa* orchid source, BYMV-SW3.2 infection caused chlorotic leaf mottle [11], while the *C. ovata* plant infected with the near-identical isolate BYMV-Devlin appeared asymptomatic. In contrast, plants infected with BYMV-BDW exhibited strong symptoms of infection. Since Koch’s postulates were not applied, it remains unclear whether the pronounced symptoms in the Bowelling-Duranillin Road population were caused by BYMV-BDW or OrMV-BDW, or both. The BYMV-BDW sequence closely resembles that of the *Gladiolus hybrida* isolate BYMV-M11 (AB079886) from Japan, which induced only mild symptoms or was asymptomatic in *Nicotiana benthamiana* and *Vicia faba* plants [23]. Because Koch’s postulates were not satisfied, it is possible that there are non-viral explanations for the symptoms observed in the Bowelling-Duranillin Road plant(s).

Like BYMV, OrMV is an aphid-transmitted potyvirus with a broad geographical and host range, but it appears to be restricted to monocotyledonous plants. It is a significant pathogen of floricultural crops, especially ornamental bulbous plants originating from Africa, such as *Iris*, *Ornithogalum*, *Lachenalia*, *Sparaxis*, *Gladiolus*, *Tritonia*, and others [24]. OrMV was first identified in Western Australia from ornamental *Iris* plants in home gardens [9]. Species of several iridaceous genera, such as *Gladiolus*, *Homeria*, *Ixia*, *Lachenalia*, *Watsonia,* and *Freesia* of southern African origin have become naturalised weeds in Western Australian bushlands [25]. It is noted that several aphid species, including *Macrosiphon euphorbiae*, *Rhopalosiphum fufiabdominalis,* and *Sitobion africanum*, infect at least some of these weeds [26]. The role of the plants as hosts for OrMV and BYMV remains untested, as does the role of aphids in spilling over exotic viruses to wild orchids. Notably, the widespread ‘South African orchid’ *Disa bracteata*, an invasive orchidaceous weed in the southwest region and wider southern Australia [27], has not been studied for its virus status or a possible role facilitating virus spillovers to indigenous orchids.

It appears there may be only one OrMV genotype present in Western Australia. The presence of close-to-identical (99%) OrMV sequences from indigenous *Diuris* orchid populations in Western Australia located 125 km from the BDW population, and from an exotic *Iris* plant from a domestic garden in Perth, Western Australia, located 230 km from the BDW population, suggests a single introduction of OrMV into Western Australia, but more extensive sampling is required to confirm this. The high genetic identity (98%) between the Western Australia OrMV isolates and an isolate from a *Lachenalia* plant in South Africa indicates that bulbous ornamental plants imported into Western Australia as garden plants, including several *Lachenalia species* [28], are sources of OrMV to the region.

More evidence is required to determine if invasive iridaceous weeds host OrMV and BYMV in Australia, and if so, the means by which these viruses are transmitted between them and indigenous plants. 

Given that the large leaves of *C. ovata* plants are on display to virus vectors throughout their lives, it was surprising that only two of the 16 populations tested were infected, and suggests that that this orchid species employs means by which it evades virus infection. 

In summary, this study demonstrated that spillover of two exotic potyviruses in two populations of the indigenous *Cryptostylis ovata* orchid occurred in Western Australia. The probable source of the OrMV genotype is one or more invasive weeds of African origin. The source of BYMV is probably leguminous crop and pasture species. *Cryptostylis ovata* is not currently listed as threatened, but nevertheless it would be wise to study the effects of virus infection on longevity and fecundity of this species. 

## Figures and Tables

**Figure 1 viruses-17-00108-f001:**
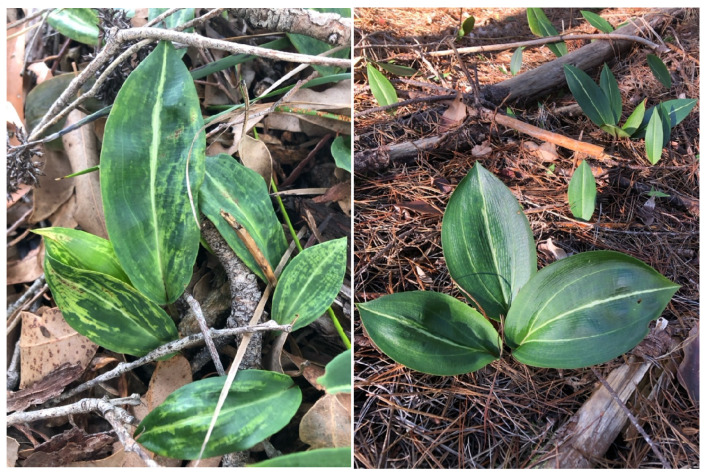
*Cryptostylis ovata* plants of the Bowelling-Duranillin Road, Wunnerberg remnant bushland population, where virus-like symptoms of yellow streaks and mosaic patterns occurred on leaves (**left**). Symptomless *C. ovata* leaves of the Coolilup population located in a *Pinus radiata* plantation (**right**).

**Figure 2 viruses-17-00108-f002:**
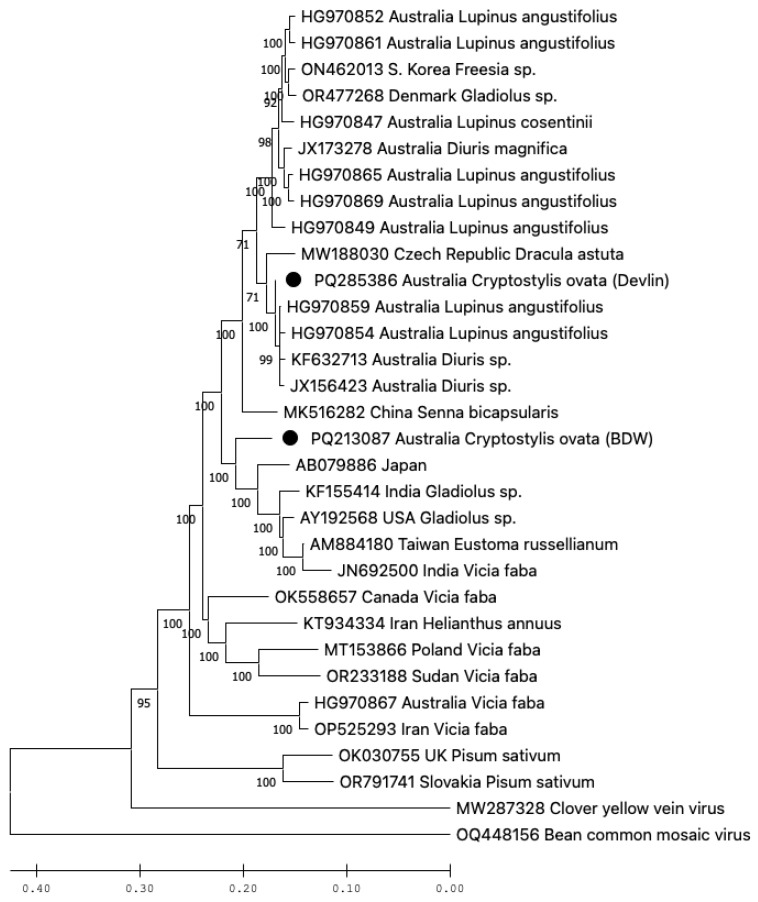
Phylogenetic tree showing genetic relationships of the large coding region (nucleotides) of bean yellow mosaic virus (BYMV) isolates Devlin and BDW from *Cryptostylis ovata* plants (indicated by black dots) compared with those of other BYMV isolates. Maximum Likelihood was used to infer evolutionary relationships. GenBank accessions are shown, followed by the country of isolation and host species for most isolates, the two exceptions being isolates of bean common mosaic virus and clover yellow vein virus, used as outgroups, where country of isolation and host species are not provided.

**Figure 3 viruses-17-00108-f003:**
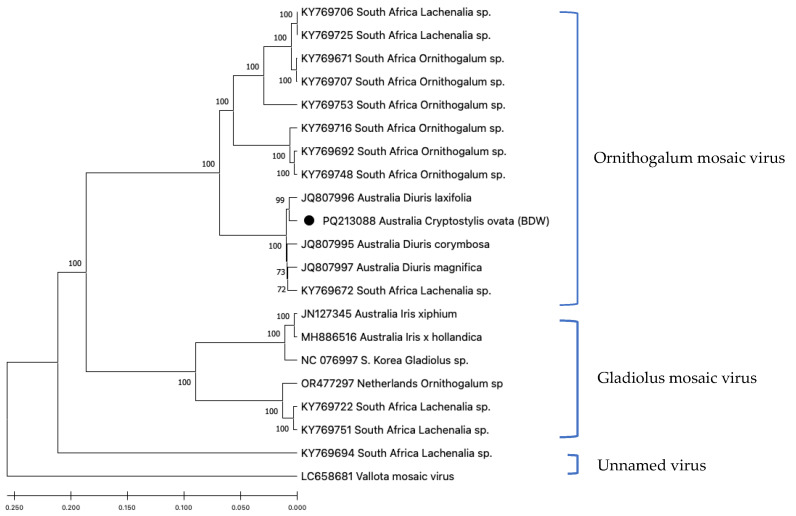
Phylogenetic tree showing genetic relationships of the genome (nucleotides) of Ornithogalum mosaic virus (OrMV) isolate BDW from *Cryptostylis ovata* plants (indicated by black dot) compared with those of other OrMV isolates. Maximum Likelihood was used to infer evolutionary relationships. GenBank accessions are shown, followed by the country of isolation and host species for most isolates. Isolates of Gladiolus mosaic virus (some isolates previously named Ornithogalum mosaic virus), an unnamed virus and an isolate of vallota mosaic virus are provided as outgroups.

**Table 1 viruses-17-00108-t001:** *Cryptostylis ovata* populations tested.

Population	Samples	Latitude	Longitude	Site Type
Abba Rd, Yoganup	5	−33.774722	115.572056	*Eucalyptus* mixed woodland
Green Hill Rd, Augusta	3	−34.29466	115.13329	*Eucalyptus* mixed woodland
Bowelling-Duranillin Rd, Wunnerberg	8	−33.43865	116.51011	Road verge remnant woodland
Johnson Rd, Bertram	6	−32.240827	115.848103	*Eucalyptus* mixed woodland
Brockman Hwy, Darradup	3	−34.151395	115.504124	Pine plantation
Inlet Dr, Denmark	1	−34.990129	117.355263	*Eucalyptus* mixed woodland
Devlin Rd, Wellesley	6	−33.223689	115.771459	Road verge remnant woodland
Forestry Rd, Uduc	1	−33.064971	115.789744	Road verge remnant woodland
Gray Rd, Boyanup	7	−33.475876	115.757871	Road verge remnant woodland
Nettleton Rd, Jarrahdale	7	−32.323770	116.070251	*Eucalyptus* mixed woodland
Lakes Rd, Dalyellup	10	−33.439839	115.608363	*Eucalyptus* mixed woodland
Coolilup, Ludlow	5	−33.608771	115.499426	Pine plantation
Windy Harbour Rd, Mt Chudalup	3	−34.765220	116.084133	*Eucalyptus* mixed woodland
Crockerup Rd, Mt Barker	1	−34.572150	117.672821	Road verge remnant woodland
Sabina Rd, Yoganup	7	−33.739220	115.499528	Pine plantation
Woods Rd, Gelorup	10	−33.430819	115.623445	Road verge remnant woodland

## Data Availability

Nucleotide sequences generated from this study are available from GenBank using the accession codes provided.

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
