# Peer review of "A Survey of Wild Indigenous *Cryptostylis ovata* Orchid Populations in Western Australia Reveals Spillover of Exotic Viruses"

_viruses, 2025, doi:10.3390/v17010108_

Round 1

Reviewer 1 Report

Comments and Suggestions for Authors

Submitted manuscript focuses on the analysis of the virome of Cryptostylis ovata, a terrestrial orchid endemic to southwestern Australia. High-throughput sequencing revealed the presence of two viruses, both exotic to Australia - bean yellow mosaic virus and ornithogalum mosaic virus. The manuscript further addresses an important topic regarding a potential spillover of harmful plant viruses from non-native invasive species to orchids indigenous to Australia. The research question is clear, and the methodology seems appropriate. However, there are areas requiring improvement, particularly in data analysis and scientific soundness of presented results. Overall, the study has potential and provides valuable insights into plant virus epidemiology.

The introduction provides adequate background, but the authors could expand on the current status of invasive iridaceous species that could serve as a reservoir of OrMV discussed in the work. The methods are generally well described; however, additional clarification on phylogenetic analyses would enhance reproducibility. The results are clearly presented, but the scientific soundness could be improved. Further details on suggested improvements are provided below. The discussion addresses key findings but could benefit from a more in-depth comparison with previous studies, if available (i.e. comparison with other cases of such spillover).

The manuscript addresses a relevant research question and adds value to the field. With improvements in mentioned areas, it could be suitable for publication.

I have several comments and suggestions regarding the submitted manuscript:

Lines 67-76 and 79-88 are identical. Please, remove the duplicity. 

Line 98: Contigs shorter than 500 nucleotides should be included in the analysis due to possible presence of viroids, the non-coding subviral pathogens that could also be the cause of observed symptoms. Since viroids do not comprise ORFs, analyses of plant viromes should not rely on protein-coding sequences only. 

Line 166: Did authors consider to perform RACE amplification to obtain a complete genomic sequence of the isolate BYMV-Devlin? 

Figures 2 and 3/Materials and Methods section: Details regarding the phylogenetic analyses should be added to the Materials and Methods section. What type of software has been used? Was the bootstrap method applied? If so, how many bootstrap replicates have been performed? 

Author Response

Submitted manuscript focuses on the analysis of the virome of Cryptostylis ovata, a terrestrial orchid endemic to southwestern Australia. High-throughput sequencing revealed the presence of two viruses, both exotic to Australia - bean yellow mosaic virus and ornithogalum mosaic virus. The manuscript further addresses an important topic regarding a potential spillover of harmful plant viruses from non-native invasive species to orchids indigenous to Australia. The research question is clear, and the methodology seems appropriate. However, there are areas requiring improvement, particularly in data analysis and scientific soundness of presented results. Overall, the study has potential and provides valuable insights into plant virus epidemiology.

The introduction provides adequate background, but the authors could expand on the current status of invasive iridaceous species that could serve as a reservoir of OrMV discussed in the work.

Response: This is a good idea, but we decided to expand on this point in the Discussion rather than the Introduction.  Because we were not aware of the potential role of invasive iridaceous weeds in spillover of exotic viruses to C. ovata orchids when we began this project, and this discovery was a significant result of this study, we reserved discussion of the role of iridaceous weeds for the Discussion section, not the Introduction. Following this suggestion, we have expanded our discussion of Iridaceous plants in the Discussion, and added new references, as follows:

The role of the plants as hosts for OrMV and BYMV remains untested, as does the role of aphids in transmitting the viruses to wild orchids. Notably, the widespread ‘South African orchid’ Disa bracteata, an invasive orchidaceous weed in the south-west region and wider southern Australia [27], has not been studied for its virus status or possible roles in virus spillovers to indigenous orchids.

More study is required to determine if invasive iridaceous weeds host OrMV and BYMV in Australia, and if so, the means by which these viruses is transmitted between them, whether by feeding aphid vectors or pollen carried by bees or the wind.

The methods are generally well described; however, additional clarification on phylogenetic analyses would enhance reproducibility.

Response: Phylogenetic methods have been added to the Materials and Methods section and removed from the figure legends.

The results are clearly presented, but the scientific soundness could be improved. Further details on suggested improvements are provided below. The discussion addresses key findings but could benefit from a more in-depth comparison with previous studies, if available (i.e. comparison with other cases of such spillover).

Response. There have been few reported spillovers of exotic viruses into the Australian orchid flora. The two published studies by Gibbs et al 2000 and Wylie et al 2013 are described in the Introduction, L53-59.

Lines 67-76 and 79-88 are identical. Please, remove the duplicity. 

Lines 79-88 have been removed.

Line 98: Contigs shorter than 500 nucleotides should be included in the analysis due to possible presence of viroids, the non-coding subviral pathogens that could also be the cause of observed symptoms. Since viroids do not comprise ORFs, analyses of plant viromes should not rely on protein-coding sequences only. 

This is an excellent suggestion. All contigs of between 101 and 500 nt were analysed for the presence of viroids using Blastn. Most of these contigs were chloroplastic DNA sequences and none matched known viroids or viruses other than the two described in the manuscript.

Line 166: Did authors consider to perform RACE amplification to obtain a complete

genomic sequence of the isolate BYMV-Devlin? 

5’RACE was not performed on any of the virus sequences obtained, including BYMV-Devlin. Although we agree it would be beneficial to perform 5’RACE on all the virus isolates, we did not have the resources for this procedure. We reasoned that since obtaining full genomes for BYMV-BDW and OrMV-BDW and over 80% of the genome of BYMV-Devlin were obtained, this information was sufficient to meet the aims of the study, i.e., 1. To identify viruses, and 2. Provide information on potential sources.    

Figures 2 and 3/Materials and Methods section: Details regarding the phylogenetic analyses should be added to the Materials and Methods section. What type of software has been used? Was the bootstrap method applied? If so, how many bootstrap replicates have been performed? 

We added this information to the last paragraph of the Materials and Methods section, as follows: Phylogenetic analysis was performed on genome sequences of viruses after alignment in ClustalW. A maximum liklihood tree was constructed within Mega11 with 1000 bootstrap replications.

Reviewer 2 Report

Comments and Suggestions for Authors

Remarks:

Section 2: The authors mention 16 populations of C. ovata plants “in southern Western Australia”. I think that for most readers, having a contour map with specific locations would be very useful.

Section 2: It is necessary to specify the other RT-PCR conditions in addition to the annealing temperature indicated in the text (60°C): elongation time, number of cycles etc.

Section 3: Have the new potyviral sequences been submitted in the GenBank (accession numbers are missing)?

Section 3: What programs were used for phylogenetic analysis?

Section 3: How did the authors define 5’-terminal sequences of the new isolates?

Figure 2: Some bootstrap values (37, 48) are too low, especially for maximum likelihood algorithm. Corresponding nodes are difficult to consider reliable; they should be collapsed.

Figure 3: What was analyzed - full-size genomes or large coding regions (see Figure 2)?

Editorial note:

Lines 71-76 and 83-88: Direct repetition of the text: “Sampling sites included a variety of habitats, ranging from remnant roadside bushland to indigenous forests and exotic Pinus radiata plantations. The aim was to collect samples from 10 individuals per population; however, in populations with fewer than 10 plants, one sample was collected from every available plant. Samples were taken regardless of visible symptoms of virus infection, such as chlorosis on young leaves, mosaic patterns, necrosis, or stunting”.

Author Response

Section 2: The authors mention 16 populations of C. ovata plants “in southern Western Australia”. I think that for most readers, having a contour map with specific locations would be very useful.

Response: After discussion between the authors, we respectfully disagree with reviewer 2 that this would add value to the manuscript because resolution would be very poor. Table 1 provides precise locations of each population as latitude and longitude figures that future researchers can access.

Section 2: It is necessary to specify the other RT-PCR conditions in addition to the annealing temperature indicated in the text (60°C): elongation time, number of cycles etc.

Response. We provided this information L107-108 as follows:

Amplification cycle conditions were melting at 95oC 10 s, annealing at 60oC 10 s, and elongation at 70oC for 20 s for 25 cycles.

Section 3: Have the new potyviral sequences been submitted in the GenBank (accession numbers are missing)?

Response: The sequences are all available on GenBank, and accession codes are shown in the figures. To clarify, we have also added these to the text as follows:

Lines 164-165. The complete genome sequence of BYMV-BDW (Genbank accession PQ213087) was 9481 nt, encoding 3056 aa, while isolate BYMV-Devlin (PQ285386)…

Line 177. The complete genome sequence of isolate OrMV-BDW was 9045 nt (PQ213088)

Section 3: What programs were used for phylogenetic analysis?

Response. This information is added at Lines 166-118 as follows: Phylogenetic analysis was performed on genome sequences of viruses after alignment in ClustalW. A maximum liklihood tree was constructed within Mega11 and the bootstrap methods with 1000 replications was used as the test of phylogeny.

Section 3: How did the authors define 5’-terminal sequences of the new isolates?

Response, as explained above for Reviewer 1, we defined the 5’-terminal regions of BYMV-BDW and OrMV-BDW using high-throughput sequencing, and the 5’-terminus of BYMV-Devlin was not determined.

Figure 2: Some bootstrap values (37, 48) are too low, especially for maximum likelihood algorithm. Corresponding nodes are difficult to consider reliable; they should be collapsed.

This was done.

Figure 3: What was analyzed - full-size genomes or large coding regions (see Figure 2)?

Response: The analysis of BYMV comprised the nucleotides of the large coding region while the analysis of OrMV sequences comprised the nucleotides of the complete genomes. This is stated in the legends of each figure as follows:

Figure 2. Phylogenetic tree showing genetic relationships of the large coding region (nucleotides) of bean yellow mosaic virus (BYMV) isolates…

Figure 3. Phylogenetic tree showing genetic relationships of the genome (nucleotides) of Ornithogalum mosaic virus (OrMV) …

Round 2

Reviewer 1 Report

Comments and Suggestions for Authors

The authors have accepted and implemented suggested revisions. I have several minor comments:

Line 98: This part should be re-written to clearly inform the reader that the analysis of contigs shorter than 500 nucleotides was also performed, according to my previous comment regarding the potential presence of viroids. 

Line 117: I suggest that the authors could specify a mathematical model used for the construction of phylogenetic tree to ensure the reproducibility of analyses. 

Author Response

Line 98: This part should be re-written to clearly inform the reader that the analysis of contigs shorter than 500 nucleotides was also performed, according to my previous comment regarding the potential presence of viroids. 

Response. This section was rewritten as follows:

'Orphan contigs were translated in six reading frames (three forward and three reverse). Contigs without open reading frames (ORFs) of at least 100 amino acid residues were discarded. For those with ORFs, nucleotide and amino acid sequences were analysed in Blastn or Blastp, respectively, for similarities to known sequences.'

Line 117: I suggest that the authors could specify a mathematical model used for the construction of phylogenetic tree to ensure the reproducibility of analyses. 

Response. This section was rewritten as follows:

'Phylogenetic analysis was performed on genome sequences of viruses after alignment in ClustalW. A maximum liklihood tree using the Tamura-Nei substitution model was constructed within Mega11 and the bootstrap method with 1000 replications was used as the test of phylogeny.'